# Leave the world(view) behind, but keep the words: The effect of conspiracism on writing

**Alessandro Miani**[1,2*], **Ines Adornetti**[3], **Daniela Altavilla**[3], **Valentina Deriu**[3], **Alessandra Chiera**[3], **Francesco Ferretti**[3]

**1** School of Psychological Science, University of Bristol, Bristol, United Kingdom, **2** Department of Psychology, University of Fribourg, Fribourg, Switzerland, **3** Cosmic Lab, Department of Philosophy, Communication and Performing Arts, Roma Tre University, Rome, Italy

* alessandro.miani@bristol.ac.uk

## Abstract

We investigate whether a stable predisposition to interpret events as the result of conspiracies—conspiracism—is associated with the spontaneous generation of conspiratorial content in writing when interpreting ambiguous information. Across two studies (*N* = 385), participants watched the apocalyptic thriller *Leave the World Behind* and wrote an essay interpreting its meaning. Each essay was rated by a Large Language Model for its conspiratorial narrative content, that is, the degree to which the text contains claims that the public is being pervasively lied to about aspects of reality, enabling some groups to enact a harmful, self-serving agenda. Contrary to our preregistered hypothesis, we did not find an association between participants' conspiracism and their essays' level of conspiratorial narrative content. Exploratory linguistic analyses revealed that conspiracism was associated with greater use of conspiracy-related vocabulary (e.g., deception, government), a disproportionate use of sophisticated words, and increased syntactic complexity. These results suggest that conspiracism may emerge more readily at the lexical level rather than through fully structured narratives. We discuss potential methodological and theoretical factors contributing to these unexpected results, including the roles of context, perceived relevance, motivation, and collective social dynamics. We also consider the possibility that conspiracism may not directly translate into conspiratorial narratives. If so, we recommend comparative research on online vs offline conspiratorial writing to clarify whether conspiracy theories emerge spontaneously from genuine beliefs or are constructed strategically, detached from genuinely held beliefs.

## Introduction

Conspiracy theories (CTs) are narratives that interpret events and aspects of reality as covert operations orchestrated by powerful actors with malevolent intentions toward an unwitting population [1–3]. Such narratives are widely endorsed [4,5],

**Data availability statement:** The data underlying the results of this study cannot be shared publicly because public data deposition was not included in the study protocol approved by the relevant Research Ethics Committee, and participant consent did not cover open data sharing. Therefore, public availability of the dataset would not be compliant with the ethical approval governing this study. The data are available from the corresponding author or from the Ethics Committee of Roma Tre University via its administrative office (Dr. Claudia Farina, claudia.farina@uniroma3.it) upon reasonable request for research purposes, subject to appropriate safeguards to protect participant confidentiality. The Ethics Committee oversees compliance with the approved study protocol and will evaluate data access requests to ensure that any data sharing is consistent with the original ethical approval and appropriate safeguards for participant confidentiality.

**Funding:** AM is supported by the Swiss National Science Foundation (SNSF, project number 214293, 'In/coherent worldviews').

**Competing interests:** The authors have declared that no competing interests exist.

persist across historical periods [6,7] and are relatively stable [8,9]. Far from trivial beliefs, endorsement of these alternative theories have mostly negative societal impacts, including decreased support for vaccination [10,11], reduced pro-environmental behavior [12,13], lower adherence to social norms [14,15], as well as increased endorsement of violence [15–17] and radicalization [18,19] (but see also the potential benefits [2] and error in dismissing CTs a priori [20]). To limit their impact, researchers need to understand the psychological underpinnings and motivations behind belief in CTs, as well as the mechanisms underlying their emergence. Yet, although substantial research has explored cognitive and social factors contributing to their endorsement [21–25], relatively little attention has been devoted to examining the content of these narratives, and virtually no work has investigated how they emerge or how individual psychological differences influence their construction.

Here, we tackle the question as to how conspiracy narratives emerge. We focus on *conspiracism*, a worldview marked by pervasive distrust of authorities and powerful actors [24,26–28] that predisposes individuals to interpret and believe events as the outcome of a conspiracy [27–29]. Note, however, that the this is a deliberate simplification. *Conspiracism* has been described and operationalized in multiple ways, including conspiracist worldview [29], monological belief system [30], generic conspiracist ideation [31], conspiracy mentality [27], conspiracy thinking [7], and conspiracist mindset [32]. Given the general and exploratory nature of this work, we use the term *conspiracism* (see, e.g., [33–35]) to indicate both the general tendency to believe in CTs *and* the possession of a conspiracy worldview. Such distinction, however, is important and will be discussed in the limitations.

Reliance on conspiratorial explanations has been associated with fundamental psychological motives, including the epistemic need for understanding and certainty, the existential drive for control and safety, and the social desire to maintain positive personal or group identities [1,2,21,22]. Note that the need-based approach, advanced by [1,2], was initially related to belief in CTs. There is however research that has linked these needs also to conspiracism (meta-analyzed in [22]), although the strength slightly varies with the way conspiracy belief is measured. Individuals confronting uncertainty or threat frequently adopt conspiratorial explanations to mitigate feelings of unpredictability and vulnerability [36,37]. Individuals high in conspiracism are generally characterized by low tolerance for ambiguity [22,38], a pronounced tendency to jump to conclusions [39], and heightened sensitivity to perceiving meaningful connections among random events [40–42]. As such, they are prone to actively seek hidden motives and ultimate truths to make sense of what happens around them [43,44]. They also frequently attribute intentionality [45] and agency to their environment [46,47]. Consequently, they are prone to build implausible associations and inferences, identifying conspiracies even where none exist [48].

**The Language of Conspiracy Theories**

Conspiracism might influence not only information processing, where uncertain or ambiguous information is interpreted through existing conspiratorial schemas, but also the way in which information is communicated. A way to explore how

conspiracism emerges in communication is to focus on the linguistic characteristics of conspiracy narratives. Research in this area is limited, but some efforts have been made. Note that narratological theories traditionally distinguish between story (the content of events), discourse or plot (their textual organization), and narration (the enunciative act that produces them; see, e.g., [49,50]). However, these distinctions are not central to the present investigation, because our analyses focus on the presence of conspiracy-related content and linguistic features rather than on formal narrative structure. Therefore, the terms *conspiratorial narratives* and *conspiratorial discourse* are used largely interchangeably throughout this article for the sake of clarity and concision.

Content analyses show that conspiratorial rhetoric is typically characterized by mistrust and refutation of mainstream perspectives [51,52], frequently targeting political opponents in positions of power [53]. On social media platforms like Reddit and Twitter, conspiratorial narratives often use words associated with crime, death, power, dominance, aggression, and deception [54–56]. Similar linguistic patterns have been found in conspiracy-related webpages compared to mainstream ones [57], suggesting a linguistic convergence between online comments and disseminated conspiracy narratives. Importantly, the use of such vocabulary predicts higher engagement and increased sharing behavior on social media [56,57]. Note, however, that these lexical patterns increase also in mainstream documents that mention conspiracies (real or theorized) in their text [57], meaning that they could indicate more of a *genre* rather than a psychological underpinning of conspiracism.

Network analyses on webpages show higher interconnectedness in conspiracy (vs mainstream) narratives via unrelated themes (e.g., Covid, 5G, Soros) [58]. A similar highly thematic heterogeneity was also found on social media such as Reddit and Twitter [59–61], where conspiracy discussions are not focused on specific events or topics and users are instead organized around diverse interests with overlapping concerns. Different from actual conspiracy (e.g., *Bridgegate*), conspiracy theories (e.g., *Pizzagate*) exhibit narrative elements that span multiple domains and rely on loosely connected relationships [62]. Given such a heterogeneity of co-occurring themes, it is not surprising that conspiratorial (vs mainstream) texts tend to show loose semantic associations [63], lower topic specificity and lexical cohesion [58] as well as lower explanatory coherence [64].

The persuasive dimension of conspiracy narratives is also crucial [65]. Experimental evidence indicates that exposure to CTs influences belief formation and behavioral intentions [13,66–68], highlighting their capacity to shape attitudes and actions. One explanation for their persuasiveness is that CTs are creatively engaging narratives [69,70], featuring spectacular, counterintuitive, and attention-grabbing rhetorical elements appealing particularly to conspiracy believers (but see [71]), who tend to be high in sensation-seeking [70] and need for uniqueness [72]. A large-scale lexical analysis confirmed that conspiratorial language employs creativity-related linguistic features such as novelty, originality, divergence, metaphoricity, and sophistication [63], features that have been associated with persuasion. Likewise, conspiracy narratives often mimic academic discourse through specialized jargon and pseudo-demonstrations [73–77], perhaps to create an illusion of rigorous analysis, expertise, and therefore increasing the perceived veracity and truthfulness of the message content. Finally, conspiracy-related (vs mainstream) content on websites tends to be longer [57,78–80], a characteristic generally associated with persuasive intent [81] (note that besides LOCO, in which differences in word count between conspiracy and mainstream texts are explicitly mentioned in the main text of the paper, the other corpora did not contain this information explicitly mentioned; here we report the correlation between word count and document-level assessment of conspiracism obtained via ChatGPT ratings in 2024: LOCO [57]: $r = .26$; DONALD [78]: $r = .11$; IRMA [79]: $r = .16$; and GERMA [80]: $r = .15$). A similar pattern was observed also on social media, where conspiracy posts on Reddit are longer [55] and tweets by conspiracy promoters are more frequent [61].

## Our contribution

Although the features of conspiracy language reviewed above consistently appear in online conspiracy discourse, a critical question remains: to what extent can we map these linguistic features to individual psychological profiles? Do

 

individual psychological differences influence the construction of conspiracy narratives? To date, no systematic study has investigated how individual differences specifically influence the emergence of conspiratorial narratives. Existing research examining contextual factors shows that exposure to conspiratorial statements decreases reliance on official information and promotes conspiratorial storylines [82,83]. However, these studies did not address the psychological precursors leaving this crucial link unexplored.

Moving beyond website-level linguistic analyses to individual-level text production is essential to clarify the psychological underpinnings of conspiratorial language. Here, we examine if *conspiracism* is associated with the interpretation and communication of narratives. In general, individuals selectively retain and transmit information that aligns with their pre-existing concerns, beliefs, and emotional states [84,85], and reinterpret complex or abstract information in ways congruent with familiar social schemas [86]. Thus, it is plausible that conspiracism may likewise lead individuals to reconstruct and communicate reality according to a conspiratorial schema, potentially fostering the emergence of conspiratorial narratives.

We hypothesize a link between conspiracism and the tendency to interpret and reconstruct ambiguous information as a result of a conspiracy. Specifically, we propose that conspiracy narratives result from top-down cognitive processes, where pre-existing beliefs and expectations guide how individuals interpret ambiguous events. We argue that cognitive biases typically associated with conspiracism lead individuals to impose conspiratorial interpretations onto ambiguous situations, particularly when epistemic needs are salient (e.g., interpreting an ambiguous situation). As a consequence, these individuals reconstruct complex or uncertain information to align it with their overarching conspiratorial frameworks.

To test this hypothesis, participants watched the apocalyptic psychological thriller *Leave the World Behind* [87] and subsequently were prompted to write an essay interpreting its meaning. This film was chosen specifically for its psychological ambiguity and the open-ended conclusion that could potentially trigger interpretations consistent with conspiracism [37]. Therefore, we measured participants' conspiracism, their essays' level of conspiracy narrative, and tested whether the two are positively associated. Our research comprises two studies: a preregistered primary study and a follow-up replication. Additionally, we performed a third exploratory analysis to examine whether participants' conspiracism level is associated with linguistic markers previously identified in online conspiratorial discourse extracted from their essays.

## Study 1

### Methods

**Procedure and participants.** The study was preregistered (https://aspredicted.org/y2kz-s7zk.pdf) and ethical approval was obtained from the Ethical Committee of Roma Tre University (Italy) in February 2024. No deviations from the approved study protocol occurred after approval was obtained (additional information regarding the ethical, cultural, and scientific considerations specific to inclusivity in global research is included in the Supporting Information S2 File). The data were collected in one day at the campus (20/05/2024). Participation was voluntary and anonymous: participants were given an identification code when they entered the classroom and received course credit as compensation. All participants provided written informed consent. Data were collected using pencil-and-paper questionnaires without computerized components and participants' responses were later transcribed by three native Italian speakers.

In accordance with our preregistered plan to collect at least 250 essays, questionnaires were distributed to over 300 students to account for potential dropouts, exclusions, and missing data. A total of 328 Italian-speaking students participated. After applying exclusion criteria—removing participants who did not consent to use their data ($N = 15$), failed attention checks ($N = 26$), or did not complete the essay task ($N = 2$)—the final sample consisted of 285 participants (221 women and 64 men; mean age = 22.25, range = 19–50).

To control for potential order effects, participants were randomly assigned to one of two groups, counterbalanced such that the questionnaire to assess participants' conspiracism was completed either before or after watching the film. Overall, the experiment lasted about 3 hours.

**The film.** Participants watched the film *Leave the World Behind* [87] and later completed the essay-writing task in which they were prompted to interpret the film by providing their personal understanding and meaning (see Prompt S1 in S1 File). We choose *Leave the World Behind* because it is not overtly conspiratorial. A conspiratorial reading is only one among several possible interpretations afforded by its narrative. We deliberately chose a non-explicitly conspiratorial film because our aim was not to test the persuasiveness of conspiratorial argumentation (see, e.g., [68]), but to examine whether people high in conspiracy mentality "fill the gaps" of an ambiguous situation, in this case, a fictional narrative.

The film is deliberately open-ended: it features unexplained phenomena, a fractured narrative timeline, and an unresolved ending that invite viewers to infer what is happening "behind the scenes". Prompting an interpretation of an overtly conspiratorial film would likely have constrained essays to a similar conspiratorial level. What interested us instead were the *additional* narrative and interpretative elements emerging from top-down cognitive processes, where pre-existing beliefs and expectations guide how individuals interpret ambiguous events.

In fact, our hypothesis is that individuals with a conspiratorial predisposition would be more likely to fill these narrative gaps with conspiratorial explanations, partly because of their higher need for structure and certainty and lower tolerance for ambiguity [22,36–38]. In this sense, the film serves as a useful tool for observing how people project conspiratorial meanings onto an ambiguous story even in the absence of an explicitly conspiratorial storyline.

Participants watched the film dubbed in Italian without subtitles. Italian dubbing is professionally standardized and ubiquitous. For most Italians a dubbed Hollywood film feels as natural as a domestic one [88]. Italians have been steeped in US screen culture since the 1950s, and recent streaming has only deepened that familiarity. Themes such as institutional breakdown, technological fragility, and post-pandemic anxiety resonate well beyond the US context. Decades of media-globalization research [89] show that such cultural proximity allows viewers to appropriate foreign narratives without losing interpretive nuance.

**Psychological measures.** To assess individual differences in conspiracism, we employed the Italian version of the 11-item Conspiracy Mentality Scale (CMS) [90], originally developed by [91]. The CMS was chosen because it measures a generic form of conspiracism like the Generic Conspiracist Beliefs scale by [31], but it is more generic (being freed from items involving specific conspiratorial themes such as the government concealing information about UFOs and aliens). Note that we also searched for the conspiracy mentality questionnaire [27] but we could not find a validated Italian translation. In our sample, the CMS demonstrated good internal consistency ($\alpha$ = .81). The CMS was originally developed to assess two specific dimensions: *healthy skepticism* (e.g., *"Many things happen without the public's knowledge"*, $\alpha$ = .76) and *conspiratorial ideation* (e.g., *"The truth is known only to a secret powerful group that actively disseminates false information or misleads the public"*, $\alpha$ = .75).

To examine the factorial structure of the CMS in our sample, we conducted a confirmatory factor analysis (CFA) using the R package *lavaan* [92]. We compared a one-factor model—treating conspiracy mentality as a single construct—with a two-factor model distinguishing skepticism from conspiratorial ideation. Model fit indices indicated that the two-factor model provided an excellent and significantly better fit compared to the one-factor model ($\chi^2$ = 119.88, $p < .001$). Although the CMS is best represented by a two-factor structure, this distinction did not substantially alter our main findings regarding the relationship between conspiracy mentality and linguistic measures. Therefore, to simplify the presentation, we report results using the single-dimension structure. Results based on the two-factor model are provided in the Appendix for completeness and transparency (as indicated in our preregistration, we also collected additional psychological measures for exploratory purposes, namely attachment style [93], translated into Italian by [94], and narcissism [95], translated into Italian by [96]; their correlations with conspiracy-related variables are reported in Table S4, see S1 File).

**Assessing conspiratorial narrative using LLMs.** As preregistered, we used Large Language Models (LLMs), specifically ChatGPT, to evaluate the conspiratorial narrative content of participants' essays. LLMs have become increasingly popular in psychological research [97,98], demonstrating their value for text classification outperforming crowd workers [99] even without task-specific training [98]. Indeed, LLMs have shown superior performance compared

to traditional dictionary-based analyses when classifying short English texts such as tweets, Reddit comments, and news headlines [100], despite some limitations in differentiating genuine inquiries and sarcastic content from authentic conspiratorial beliefs [101]. Given these considerations, we implemented a two-step validation procedure: first, we assessed LLMs' reliability in identifying conspiracy-related content in English texts and second, we validated its performance specifically on Italian-language texts.

To evaluate LLMs' effectiveness to label English texts as conspiracy, we relied on textual material gathered from two corpora: LOCO (the Language Of COnspiracy corpus [57]) and DONALD (the 2M-document Dataset Of News Articles for studying the Language of Dubious information [78]). LOCO is an 88-million-word corpus comprising 23,937 conspiracy and 72,806 mainstream texts focused on 47 well-known themes that have generated CTs (e.g., Princess Diana's death, the Moon landing, the 9/11 terrorist attacks). DONALD consists of approximately 2 billion words across 2,173,172 news articles—217,703 of which originated from conspiracy websites—covering 172 politically polarized topics (e.g., abortion, immigration, civil rights, climate change).

We prompted ChatGPT by OpenAI—one of the most popular and cost-effective LLMs at the time of writing (in July 2025, ChatGPT's `GPT-4o-mini` model cost $0.15 + $0.6 (input + output) per one million tokens, compared to Claude `claude-3-5-sonnet` that costs $3 + $15 per one million tokens)—to assign a fine-grained conspiratorial language score to each document on a 0–10 Likert scale (from *non-conspiratorial language* [0] to *highly conspiratorial language* [10]). Results showed that conspiracy-related documents received significantly higher scores ($M = 6.93$, $SD = 2.61$) compared to mainstream texts ($M = 1.58$, $SD = 1.80$; $\beta = 1.741$, $SE = .005$, $p < .001$). Approximately 77% of conspiracy-related documents were rated by ChatGPT above 5, whereas 95% of mainstream documents were rated below 5.

To examine generalizability beyond LOCO's prominent conspiracy-related themes, we selected from DONALD a subset of 144,000 documents (12,000 per ideological domain, each between 500 and 1,500 words). We then validated ChatGPT's ratings by computing their correlation with website-level reliability scores obtained from [102]. This analysis revealed a strong negative correlation ($r = -.67$ [−.693, −.637], $p < .001$), indicating that texts from less reliable sources received higher conspiratorial scores.

While ChatGPT demonstrated reliability for English texts, its accuracy with Italian-language texts required further validation. We therefore tested its performance using IRMA [79], a corpus of over 600,000 Italian news articles (approximately 335 million words) from 56 websites categorized as unreliable by professional fact-checkers. We selected a subset of 2,714 documents from IRMA (up to 50 per website where available, each under 500 words), which likely included a mix of conspiratorial and non-conspiratorial texts. Each document was scored by ChatGPT for conspiratorial content (again using a 0–10 Likert scale).

To establish a human-coded benchmark, we randomly selected 180 documents spanning the entire range of ChatGPT scores, which were independently evaluated by three native Italian speakers (60 documents per rater) using the same scoring method. We compared ChatGPT ratings (using the `gpt-4o-mini` model) with those of another LLM, Claude by Anthropic (using the `claude-3-5-sonnet` model). The correlation between human ratings and LLMs's ratings was overall robust, indicating strong agreement, although Claude ($r_{176} = .701$, [.617, .769], $p < .001$) performed slightly better than ChatGPT ($r_{178} = .676$, [.588, .748], $p < .001$; but note that the 95% confidence intervals overlap).

Based on these validation steps, we concluded that LLMs provide reliable evaluations of conspiracism and that Claude performed slightly better than ChatGPT for Italian texts. Thus, we employed Prompt S2 (see S1 File) to score participants' essays via Claude (note that the use of Claude deviates from our preregistered use of ChatGPT), adopting the definition of CTs provided by [3]. Note that before being acknowledged with Claude, we used ChatGPT to evaluated 108 items previously rated by human coders—MTurk workers and master's students—for conspiratorial content. The definition of CTs provided by [3] used as prompt for ChatGPT yielded the highest predictive accuracy ($R^2 = .644$) compared to three alternative definitions (ChatGPT's own definition: $R^2 = .639$; [2]: $R^2 = .595$; and [21]: $R^2 = .637$).

## Results

We started by assessing whether prior exposure to the film influenced key variables in our study. No significant differences emerged in conspiracism levels between participants who had previously watched the film ($N = 43$; $t_{53.79} = 1.43$, $p = .158$) and those who had not. Similarly, prior exposure did not significantly influence the conspiratorial content of participants' narratives ($t_{52.49} = -.59$, $p = .561$). Next, we examined whether completing the questionnaire before ($N = 161$) or after ($N = 124$) watching the film affected conspiracism or the conspiratorial content of essays. Again, there were no significant differences between the two groups, neither in terms of conspiracism ($t_{259.52} = .06$, $p = .949$) nor regarding the conspiratorial content of essays ($t_{277.91} = -.20$, $p = .845$).

We had preregistered our hypothesis that participants with higher conspiracism would interpret ambiguous information in a conspiratorial manner, as reflected in the content of their essays. Contrary to our expectations, however, conspiracism was not significantly correlated with the conspiratorial content of essays ($r_{283} = .06$, $[-.06, .18]$, $p = .305$). Since the CMS has two-factor structure, we also ran an additional regression treating the two CMS subdimensions (conspiratorial ideation and healthy skepticism) as separate predictors of the conspiratorial narrative scores. Again, there were no significant effects of either subscale (ideation: $\beta = .05$, $[-.08, .19]$, $SE = .07$, $t = .76$, $p = .446$; skepticism: $\beta = .02$, $[-.12, .15]$, $SE = .07$, $t = .23$, $p = .818$), indicating that the null finding observed in our original analysis does not depend on the aggregation of the two subcomponents of the CMS.

## Discussion

One possible explanation for these unexpected results may relate to our measurement of conspiracism (i.e., the Conspiracy Mentality Scale, CMS) and the sociopolitical context in which it was administered (Italy). Specifically, the CMS might not fully capture conspiracism per se, but instead measure related constructs such as political knowledge or institutional skepticism.

In reconsidering the CMS items within the Italian sociopolitical context, several items do not appear inherently conspiratorial. Given Italy's record on press freedom and political transparency [103,104], some statements included in the scale may reflect political awareness or legitimate skepticism rather than conspiracism (e.g., *"There are people who don't want the truth to come out"* or *"Many things happen without the public's knowledge"*). This interpretation aligns with cross-national research conducted in 26 countries by [105], which identified excessive cross-country variability in one item from the Conspiracy Mentality Questionnaire (CMQ; [27]): *"I think that politicians usually do not tell us the true motives for their decisions"*. Such variability likely reflects differing levels of institutional trust rather than actual conspiratorial mentality. Consequently, items that appear conspiratorial in transparent democracies might simply reflect legitimate political skepticism in less transparent environments.

Moreover, the Italian version of the CMS employed in this study [90] was validated within the Italian-speaking region of Switzerland—a context politically and culturally different from Italy. Although validation included assessments related to COVID-19 measures, it did not involve comprehensive convergent validation against well-established conspiracy belief scales such as the CMQ [27] or the Generic Conspiracist Beliefs scale (GCB; [31]). Thus, it remains uncertain whether the CMS effectively captures conspiracism within the Italian (as opposed to Swiss) context.

To address these methodological concerns, we conducted a follow-up replication study employing the Generic Conspiracist Beliefs scale (GCB), a measure specifically designed to capture generalized conspiratorial beliefs.

## Study 2

To address methodological concerns identified in Study 1, we conducted a replication with a single modification: instead of the CMS [90], we used the Italian version of the 15-item Generic Conspiracist Beliefs scale (GCB; translated by [106]), originally developed by [31]. The GCB assesses generalized conspiratorial beliefs explicitly referencing specific

conspirators, such as governments or scientists (e.g., *"The government is involved in the murder of innocent citizens and/ or well-known public figures, and keeps this a secret"*). Given its explicit identification of conspirators, the GCB might offer improved sensitivity for conspiracism in our Italian sample and predictive validity to the emergence of conspiratorial narrative relative to the CMS.

## Methods

All aspects of Study 2—including measures (apart from the conspiracy scale), recruitment strategy, and experimental tasks—remained identical to those in Study 1. The study took place over four days (between 30/10/2024 and 06/11/2024). A total of 117 Italian-speaking undergraduate and graduate students participated. After exclusions—lack of consent to use data ($N = 3$), failing attention checks ($N = 14$), or incomplete essays ($N = 6$)—the final sample consisted of 100 participants (66 women and 34 men; mean age = 24.56, range = 18–59). As in Study 1, we also collected exploratory measures of attachment style and narcissism; their correlations with conspiracy-related variables are reported in Supporting Information (see Table S5 in S1 File).

## Results

Consistent with Study 1, prior exposure to the film did not significantly influence key variables. Participants who had already seen the film ($N = 16$) did not differ significantly in the GCB scores from those who had not ($t_{24.01} = -1.06$, $p = .301$), and prior watching similarly did not affect the conspiratorial content of their narratives ($t_{22.41} = .23$, $p = .822$). Additionally, no significant order effects emerged regarding whether participants completed the questionnaire before ($N = 48$) or after ($N = 52$) watching the film, neither for the GCB scores ($t_{94.63} = 1.33$, $p = .186$) nor for conspiratorial content in the essays ($t_{97.71} = -1.11$, $p = .271$).

Despite employing a different conspiracism scale (GCB rather than CMS), we again observed no significant correlation between participants' conspiracism and the conspiratorial content of their narratives ($r_{98} = .11$, $[-.09, .30]$, $p = .262$). As in Study 1, we ran an additional regression treating the five subdimensions of the GCB as separate predictors of the conspiratorial narrative scores. Again, there were no significant effects of individual subscale except one (Government malfeasance: $\beta = .31$, $[.04, .58]$, $SE = .14$, $t = 2.25$, $p = .027$; Malevolent global conspiracies: $\beta = -.04$, $[-.29, .21]$, $SE = .13$, $t = -.31$, $p = .754$; Extraterrestrial cover-up: $\beta = -.24$, $[-.47, -.02]$, $SE = .11$, $t = -2.15$, $p = .034$; Personal wellbeing: $\beta = .10$, $[-.18, .39]$, $SE = .14$, $t = .72$, $p = .473$; Control of information: $\beta = -.05$, $[-.31, .21]$, $SE = .13$, $t = -.38$, $p = .705$).

## Discussion

In Study 2, we attempted to address potential limitations of Study 1 by implementing an alternative and more explicit measure of conspiracy beliefs (GCB). However, contrary to our hypotheses, we still did not find a significant association between conspiracism and the production of conspiratorial narratives, although the effect sizes were somehow larger. These findings mitigated our concerns related to the CMS's appropriateness in the Italian context, as the anticipated relationship similarly failed to emerge with the GCB. It seems unlikely that both scales systematically failed to capture conspiracism, thus suggesting that the lack of association observed is not simply due to inadequate measurement. One interpretation of these findings is that the hypothesized relationship between conspiracism and conspiratorial narrative level genuinely does not exist, at least within the conditions of this study. Consequently, the LLM might have accurately detected the absence of explicitly conspiratorial narratives, reflecting a real absence of this link in spontaneous text production.

However, if this relationship does indeed exist but went undetected, other sources of measurement error should be considered. For instance, it is possible that the LLM reliably identifies conspiratorial narratives only when these narratives are clearly structured and cohesive. Participants holding conspiratorial beliefs might have expressed their ideas implicitly, fragmentarily, or incoherently—forms of expression that the LLM may have failed to classify as conspiratorial. Therefore, the

reliance of our textual measure on narrative cohesion may represent a critical limitation. It is possible that alternative methods could identify conspiracism in writing without relying on well-structured texts, such as dictionary-based methods (bag-of-words analyses), which could capture conspiratorial elements in less structured texts without requiring narrative coherence.

## Study 3: Exploring the individual language of conspiracism

Building upon the limitations discussed in Studies 1 and 2, we examined whether participants' essays exhibited linguistic patterns previously associated with conspiratorial online discourse as reviewed above (see The Language of Conspiracy Theories). Specifically, we investigated linguistic features that provide an alternative assessment of textual conspiracism—i.e., the conspiratorial lexicon—along with measures related to textual quality, such as linguistic (lexical and syntactic) sophistication and textual cohesion.

These exploratory analyses served two primary aims. First, given that this study is—to our knowledge—the first to examine individual-level linguistic markers of conspiracism, we explored whether these theoretically grounded linguistic measures correlate with participants' self-reported level of conspiracism. This allowed us to preliminarily assess whether linguistic patterns observed in online conspiracy discourse also manifest in offline, individual text production as a function of conspiracism. Second, because there is some evidence that LLMs rely on narrative coherence and stylistic cues [107,108] these measures serve also to assess LLM's evaluations, and whether they are sensitive to textual quality.

Additional exploratory analyses using the Linguistic Inquiry and Word Count software (LIWC; [109], translated in Italian by [110]) are reported in S1 File (Table S6), including the correlation coefficients between conspiracism and conspiratorial content (lexical and narrative) identified in participants' essays.

### Material

Because participants from Studies 1 and 2 were provided the same essay prompt (i.e., interpreting the film *Leave the World Behind*, see Prompt S1 in S1 File), and data collection procedures were consistent across both studies, we aggregated essays from both studies into a single corpus to increase statistical power (total $N = 385$).

To create a unified measure of conspiracism, we combined the Conspiracy Mentality Scale (CMS) used in Study 1 and the Generic Conspiracist Beliefs scale (GCB) from Study 2 into a single composite index. This approach follows prior research practices where measures from separate studies, employing Likert scales with different response ranges, are standardized and aggregated [33]. Responses from each scale were rescaled to a common 0–1 range (0 = *strongly disagree*, 1 = *strongly agree*, with an *unsure* midpoint at .50) using the formula $z = \frac{x - \min(L)}{\max(L) - \min(L)}$ where $z$ and $x$ denote the rescaled and original response values on scale $L$, respectively, and $\min(L)$ and $\max(L)$ represent the theoretical minimum and maximum response values of the Likert scale.

No significant difference in conspiracism emerged between participants in Study 1 and Study 2 ($t_{139.48} = 1.49$, $p = .14$). Participants in Study 2, however, were older on average ($t_{160.84} = -4.51$, $p < .001$) and had higher educational levels ($t_{163.39} = -8.34$, $p < .001$) compared to those in Study 1.

### Linguistic indices

**Conspiratorial lexicon (dictionary).** Given the lexical convergence of conspiratorial discourse observed across social media [54,55] and online texts [57], we explored whether participants' essays similarly reflected a conspiratorial lexicon corresponding to their level of conspiracism. To this end, as an alternative to LLM, we employed a dictionary-based approach for assessing lexical conspiracism. A dictionary is a predefined list of words associated with a specific psychological construct or phenomenon. Widely used in psychological research [98,109,111], this method consists in identifying and counting occurrences of words in a text that match entries in the dictionary. To ensure comparability across documents of varying lengths, the frequency of matches is expressed as a ratio ranging from 0 (no dictionary matches) to 1 (all words matched), allowing for standardized comparisons between texts.

We are currently developing a data-driven dictionary of conspiracism in English [112]. Unlike LLMs—which may depend on narrative structure and textual coherence—the dictionary method quantifies conspiratorial content solely based on word frequency, independent of word order or narrative cohesion [98]. The dictionary used in this study comprises 300 English lemmas derived from the topic-matched structure of LOCO [57]. Specifically, by contrasting conspiracy with mainstream texts within each seed topic in LOCO (e.g., the death of Princess Diana), we employed a term frequency–inverse document frequency (TF-IDF) approach to isolate conspiracy-specific terms while filtering out topic-specific terms treated as stop words (e.g., *car crash* and *Paris* in the context of the death of Princess Diana). As such, this method effectively identifies lexical markers of conspiracism that generalize across diverse topics. To validate our dictionary, we applied it to the subset of 144,000 documents from DONALD previously scored for conspiratorial content using ChatGPT (as described above). The dictionary-derived scores significantly correlated with ChatGPT ratings ($r = .60$, [.597, .605], $p < .001$).

For this work, the dictionary was translated into Italian using ChatGPT (see Prompt S3 in S1 File). The complete list of Italian dictionary terms is provided in the S1 File (see Section S2). The Italian dictionary was then applied to the lemmatized version of our combined corpus from Studies 1 and 2. Lemmatization was performed using the R package *spacyr* [113], a wrapper for the Python library *spaCy* [114], which provides accurate extraction of grammatical structure and prevents potential mapping errors (e.g., distinguishing between the Italian noun *mondo* [world] and the verb *mondare* [cleanse]). We selected the pretrained model `it_core_news_lg` due to its superior performance among available Italian models in spaCy.

After lemmatization, each text was reconstructed by concatenating its lemmatized tokens. Tokenization was subsequently performed using the R package *quanteda* [115], removing punctuation, symbols, and numbers. Two punctuation marks (`?` and `!`) were retained, as they appeared in the conspiratorial dictionary. Compound words were also combined (e.g., *sito web* became *sito_web* [website]). Finally, the dictionary was applied to the processed texts, computing the proportion of conspiratorial words in each document. The resulting scores ranged theoretically from 0 (no matches) to 1 (all words matched).

**Sophistication.** Conspiracy narratives often feature high levels of linguistic sophistication, characterized by complex syntax and specialized vocabulary. Linguistic sophistication can serve persuasive goals [116,117], strategically creating impressions of truthfulness, competence, and expertise within conspiratorial discourse [73]. Empirical research suggests that conspiracy-related texts generally exhibit greater lexical and syntactic complexity compared to non-conspiracy texts [63,118,119]. It is thus plausible to expect similar linguistic sophistication in offline contexts, such as in the essays analyzed here, as a function of individual levels of conspiracism. However, considering that conspiracy believers typically demonstrate lower verbal intelligence scores [120] and tend to have less formal education [121], linguistic sophistication may not straightforwardly correlate with conspiratorial belief. In the following, we describe how we operationalize linguistic sophistication using two complementary dimensions: syntactic and lexical sophistication.

Note that beyond syntactic and lexical sophistication, previous research has also examined how semantic and pragmatic features (e.g., counterfactuals, hypotheticals, and evaluative language) contribute to conspiracy discourse, particularly in relation to refutational reasoning [122]. While these aspects fall outside the scope of our structurally focused analysis, we acknowledge their relevance and encourage future work to incorporate verb semantics and discourse-level patterns for a more comprehensive understanding of conspiratorial rhetoric.

**Syntactic Sophistication.** To measure syntactic sophistication, we extracted three key metrics from the parsed texts of participants' essays. First, we calculated the *average sentence length* by counting the number of tokens per sentence and computing the mean sentence length for each participant, with longer sentences indicating greater syntactic complexity. Second, we measured the *average clauses per sentence*, defined by counting all verbs (including auxiliaries) as proxies for clauses and computing the average number of clauses per sentence for each participant; a higher average suggests more complex clausal embedding. Third, we determined the *average dependency tree depth* by calculating the

maximum depth of dependency structures in each sentence (i.e., the absolute positional difference between each token and its syntactic head) and averaging these maximum depths across all sentences per participant. Greater tree depth means greater hierarchical syntactic complexity.

To create a unified measure of syntactic sophistication, we combined these three metrics using Principal Component Analysis. The first principal component accounted for approximately 91% of the variance and served as our composite measure of syntactic sophistication, effectively capturing multiple aspects of syntactic complexity in a single score. We validated our syntactic sophistication measure using a synthetic corpus specifically created by ChatGPT ($N$ = 50 documents) containing texts of explicitly high or low syntactic complexity (see Prompt S4 in S1 File) As expected, the high-complexity texts scored higher in syntactic sophistication compared to low-complexity texts ($t_{44.61}$ = 9.18, $p < .001$, Cohen's $d$ = 2.60).

We applied our syntactic sophistication index to the LOCO corpus. We found that, consistent with prior findings on social media [118,119], conspiracy documents showed greater syntactic complexity than mainstream texts ($\beta$ = .07, [.05, .08], $SE$ = .01, $t$ = 8.34, $p < .001$; using a multilevel regression models via the *lme4* and *lmerTest* packages in R [123,124], including random intercepts for $k$ = 200 topics, thus ensuring within-topic comparisons).

**Lexical Sophistication.** To our knowledge, no existing word norms directly quantify lexical sophistication. Instead, researchers have often inferred lexical sophistication indirectly via metrics such as processing time that is known to correlate with age of acquisition (AoA; [125]), word frequency [126], and word length [127]. To address this gap, we generated lexical sophistication norms using LLMs, which have shown promising results in approximating human psycholinguistic judgments [128]. Specifically, we extracted a list of Italian lemmas and their part-of-speech tags from the IRMA corpus [79], and prompted ChatGPT to estimate the complexity of each word analogously to AoA ratings. We prompted ChatGPT to assess how sophisticated a word would appear to a native Italian speaker (see Prompt S5 in S1 File). We anchored the ratings in AoA norms because they reflect the developmental trajectory of lexical acquisition: early-acquired words (e.g., *mum*, *food*) are typically perceived as less complex, whereas later-acquired or specialized terms (e.g., *syphilis*, *kerosene*, *neurotic*) are typically perceived as more complex. AoA norms have previously been used as a proxy for linguistic sophistication [129] and have captured sophistication in nominal compounds in conspiracy texts [63].

We obtained sophistication ratings for 190,507 Italian lemmas. These ratings showed strong convergent validity with the existing Italian AoA dataset [130], yielding a correlation of $r$ = .64 [.611, .665], $p < .001$ across the 1,912 overlapping lemmas. Using the same prompt, we also generated complexity norms for English and German (each with $N > 100,000$ lemmas). These norms similarly correlated well with established AoA datasets (in English: $r$ = .74, based on 28,000 words from [125]; in German: $r$ = .71, based on 2,958 words from [131]). Note that while these correlations are encouraging, the cognitive mechanisms underlying LLM-generated norms remain opaque. Future work should explore whether these norms exhibit the same cross-linguistic consistency as human AoA ratings [132]. Applying the English LLM-based norms to the LOCO corpus, we found that conspiracy texts exhibited significantly higher lexical sophistication than mainstream texts ($\beta$ = .07, [.06, .09], $SE$ = .01, $t$ = 9.76, $p < .001$; using multilevel regression with a random intercept for $k$ = 200 topics).

Here, for participant essays, we computed lexical sophistication by mapping the generated Italian AoA ratings onto each lemmatized text and calculating the mean per participant.

**Megalalia.** It is possible that linguistic sophistication in conspiratorial discourse is not uniformly distributed. Rather, conspiracy authors might selectively employ specialized terminology across diverse and unrelated contexts [63], even within texts that are otherwise linguistically simple. In this view, conspiratorial linguistic sophistication could be defined by an uneven distribution of sophisticated terms within the discourse.

We are currently developing a metric designed specifically to quantify the disproportionate use of sophisticated vocabulary [77]. We label this phenomenon as *megalalia* (from Greek *mega* = "big" and *lalia* = "speech"), referring to the inappropriate use of highly sophisticated vocabulary without regard for context or necessity. This linguistic behavior often serves the strategic purpose of creating an illusion of intelligence, credibility, or authority [73,117].

 

The concept of megalalia relies on the assumption that sophisticated words, when used out of context or unnecessarily, should stand out as disproportionately complex relative to the surrounding lexical context. To quantify megalalia, we applied the Gini coefficient to the distribution of lexical sophistication scores within each text. Originally developed to measure economic inequality, the Gini coefficient quantifies disparity within a distribution. Applied to lexical sophistication, a Gini coefficient of zero would reflect complete uniformity indicating equal lexical sophistication across the entire text (e.g., *pizza*, *pasta*, *table*), whereas values approaching one would suggest high inequality, i.e., a small subset of sophisticated words contrasts sharply with otherwise simpler vocabulary (e.g., *pizza*, *pasta*, *orbitofrontal cortex*). This pattern of selectively inserting sophisticated terminology could reflect attempts to exaggerate perceived expertise or credibility, characteristic of conspiratorial arguments [73]. Testing this measure on the LOCO corpus revealed significantly higher levels of megalalia in conspiracy texts compared to mainstream texts ($\beta = .27$, [.25, .28], $SE = .01$, $t = 32.78$, $p < .001$; using the same multilevel model structure as described previously).

**Cohesion.** A large-scale network analysis of conspiracy texts using LOCO revealed that conspiratorial discourse exhibits lower textual cohesion compared to mainstream texts, characterized by frequent co-occurrences of unrelated topics [58]. Similar fragmented and associative thought patterns have been observed among individuals on the schizophrenia spectrum [133–135], which partially overlaps with conspiracism (schizotypy is the strongest psychopathological predictor of conspiracism [24]). Based on these observations, we examined whether textual cohesion in participants' essays negatively correlated with their level of conspiracism. Because previous assessments of cohesion in conspiracy-related texts were conducted exclusively in English [58], here, we developed a novel approach to measure textual cohesion in other-than-English languages.

We quantified textual cohesion as the semantic similarity between consecutive sentences within each document. To this goal, we employed word embeddings from the Italian pre-trained `fastText` model [136,137] (300 dimensions; https://fasttext.cc/docs/en/crawl-vectors.html), which offer distinct advantages for morphologically rich languages like Italian by leveraging subword information [138]. Each sentence was represented as the mean vector of its constituent word embeddings, after removing stopwords (list of 279 stopwords from *quanteda*; [115]) and punctuation. Sequential textual cohesion scores were then computed by calculating the cosine similarity between embeddings of adjacent sentences within each text. Higher sequential cohesion values indicate greater semantic similarity between consecutive sentences, reflecting more coherent and structured discourse.

To validate our cohesion measure, we adopted a procedure similar to that used in previous research [58]. We compared cohesion scores of original texts to their scrambled counterparts, in which sentence order was randomized while preserving word order within sentences. Specifically, we selected a random sample of 100 documents, each containing at least 20 sentences, from the IRMA corpus [79]. As expected, the scrambled documents exhibited significantly lower sequential cohesion than the original documents ($t_{186.51} = 5.63$, $p < .001$, $d = .80$), confirming our metric's sensitivity to disruptions in textual continuity. Further validation using the English fastText model was performed on a synthetic dataset ($N = 770$) from prior work (see section *Testing cohesion metrics* in [58]), yielding similarly results ($t_{723.91} = 16.19$, $p < .001$, $d = 1.17$).

## Results

A correlation matrix summarizing the relationships among participants' education level, conspiracism, and textual variables extracted from their essays is presented in Table 1. There were no significant differences between Study 1 and Study 2 regarding either conspiratorial narratives (assessed via the LLM; $t_{172.39} = -.02$, $p = .982$) or conspiratorial lexicon (assessed via the dictionary; $t_{213.86} = 1.66$, $p = .098$). Participants' conspiracism scores were not correlated with conspiratorial narrative scores ($r_{383} = .08$, [−.02, .18], $p = .134$; Study 1: $r_{283} = .06$, [−.06, .18], $p = .305$, Study 2: $r_{98} = .11$, [−.09, .30], $p = .262$), but they were positively correlated with the use of conspiratorial lexicon ($r_{383} = .19$, [.09, .29], $p < .001$; Study 1: $r_{283} = .18$, [.06, .29],

**Table 1. Correlation matrix of exploratory textual variables (Studies 1 and 2 aggregated).**

| | 1. | 2. | 3. | 4. | 5. | 6. | 7. | 8. |
|---|---|---|---|---|---|---|---|---|
| 1. Conspiracism | | | | | | | | |
| 2. CT narrative | .08 | | | | | | | |
| 3. CT lexicon | .19*** | .30*** | | | | | | |
| 4. Sophistication (L) | −.17** | .10* | −.15** | | | | | |
| 5. Sophistication (S) | .15** | .06 | .01 | −.12* | | | | |
| 6. Cohesion | .06 | .06 | .03 | −.04 | .65*** | | | |
| 7. Megalalia | .10* | −.06 | .13* | −.20*** | −.09 | −.10 | | |
| 8. Word Count | −.01 | .08 | −.12* | −.02 | .07 | .04 | .04 | |
| 9. Education | −.23*** | .02 | −.17*** | .17*** | −.03 | .02 | −.07 | .13** |

Note. $N = 385$. Conspiracism scores were obtained by rescaling CMS and GCB responses to a 0–1 range. *CT Narrative*: Conspiratorial narrative scores based on LLM assessments; *CT Lexicon*: Conspiratorial lexicon scores based on dictionary assessments; Sophistication: L = lexical, S = syntactic.

$p = .003$, Study 2: $r_{98} = .23$, [.04, .41], $p = .02$). This pattern suggests that individuals with stronger conspiracism tend to use conspiracy-related vocabulary, even if they do not produce clearly identifiable conspiratorial narratives. Scatterplots illustrating these relationships are provided in Fig 1.

As we mentioned, reliance on Claude (vs ChatGPT) represents a deviation from our preregistration. For transparency, we report the correlation between ChatGPT assessment and conspiracism level: $r_{383} = .01$, [−.09, .11], $p = .821$. Note that we also tested an alternative prompt to gather conspiracy scores from LLMs (see Section S3 in S1 File for details). Instead of using the definition of [3], we prompted Claude to rate in each essay different dimensions theoretically associated with conspiracism (e.g., deception, social relevance, etc, [2,139]). Scores following this new prompt were positively associated with Prompt S2 (in S1 File) based on the definition of [3] ($r_{383} = .67$, [.62, .72], $p < .001$), but they were not associated with conspiracism ($r_{383} = −.01$, [−.11, .09], $p = .9$).

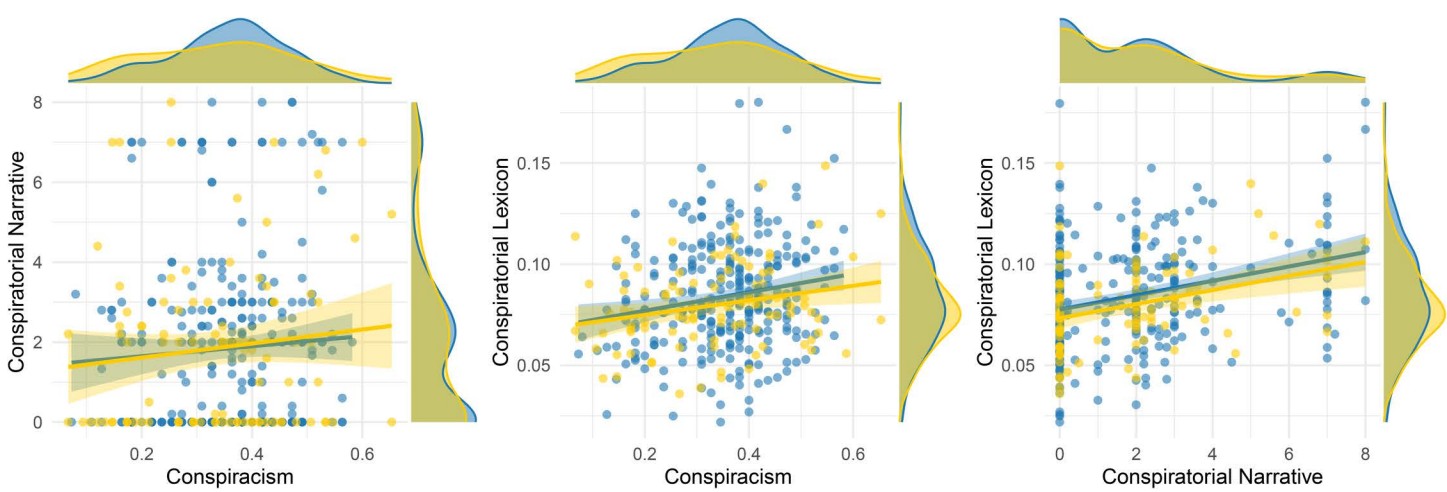

**Fig 1. Scatterplots and distributions of conspiracy-related variables.** Participants' conspiracism was assessed using the CMS and GCB. Essays' conspiratorial narrative scores were evaluated via LLM, while conspiratorial lexicon scores were computed using the dictionary method. Data from Study1 (■) and Study2 (■) are combined ($N = 385$).

Participants' level of conspiracism was also positively—albeit weakly—associated with megalalia ($r_{383} = .10$, [.00, .20], $p = .045$), indicating that individuals with stronger conspiracism were more likely to employ sophisticated vocabulary disproportionately within otherwise simple texts, possibly reflecting strategic impression management. Megalalia was additionally correlated with conspiratorial lexicon ($r_{383} = .13$, [.03, .23], $p = .011$) but was not related to conspiratorial narratives ($r_{383} = -.06$, [−.16, .04], $p = .254$).

We next examined whether textual quality predicted conspiratorial narratives or lexicon. The level of conspiratorial narrative assessed via LLM correlated positively and significantly with lexical sophistication ($r_{383} = .10$, [.00, .20], $p = .045$), but not with syntactic sophistication ($r_{383} = .06$, [−.04, .16], $p = .255$), textual cohesion ($r_{380} = .06$, [−.04, .16], $p = .258$), and word count ($r_{383} = .08$, [−.02, .18], $p = .124$)—although these relationships were positive. Conversely, conspiratorial lexicon assessed via dictionary was negatively correlated with lexical sophistication ($r_{383} = -.15$, [−.25, −.05], $p = .003$) and word count ($r_{383} = -.12$, [−.22, −.02], $p = .019$), and was not associated with syntactic sophistication ($r_{383} = .01$, [−.09, .11], $p = .833$) nor textual cohesion ($r_{380} = .03$, [−.08, .13], $p = .621$).

Participants' educational levels correlated positively with lexical sophistication ($r_{381} = .17$, [.07, .27], $p < .001$) and word count ($r_{381} = .13$, [.03, .23], $p = .01$), but, surprisingly, not with syntactic sophistication ($r_{381} = -.03$, [−.13, .07], $p = .569$) or cohesion ($r_{378} = .02$, [−.09, .12], $p = .764$). Thus, higher education levels are associated with richer vocabulary usage and longer essays, but not necessarily with more complex syntax or greater textual cohesion. Education was negatively associated with both conspiracism ($r_{381} = -.23$, [−.32, −.13], $p < .001$) and conspiratorial lexicon use ($r_{381} = -.17$, [−.27, −.07], $p < .001$), but not with conspiratorial narrative ($r_{381} = .02$, [−.08, .12], $p = .675$). This indicates that more educated individuals reported lower conspiracism and used fewer conspiracy-related terms, but were not more or less likely to produce a conspiratorial narrative.

To evaluate whether the textual quality of essays could affect the LLM's assessments of conspiratorial levels (see, e.g., [107,108]), we conducted a series of regressions predicting essays' conspiratorial scores from participants' conspiracism, adding textual quality as moderator (for a thorough description of the methods with results, see Section S4 in S1 File). A significant interaction could indicate that the relationship between participants' conspiracism and their essays is moderated by the quality of the texts. Note that because we preregistered the use of ChatGPT, but diverged from the preregistration because we discovered that Claude yielded slightly better results, we performed these tests for both ChatGPT and Claude along with our dictionary. We ran nine regression models in which we predict the essay conspiratorial scores (evaluated with dictionary, ChatGPT, and Claude) from conspiracism interacting with measures of textual quality (syntactic sophistication, lexical sophistication, and cohesion). Overall, conspiracy narratives evaluated via LLMs were consistently and positively associated with textual qualities ($\beta$s > .10, $p$s < .056) but not with conspiracism (|$\beta$s| < .04, $p$s > .05). When essays' conspiracy level was assessed via dictionary, the reverse pattern was found, with a positive and consistent association with conspiracism ($\beta$s > .17, $p$s < .001) but not with textual quality, which in fact showed a negative association ($\beta = -12$, $p < .024$). As for the interactions, on average, the absolute coefficients of the models for LLM-based assessments were larger than those for the dictionary-based assessment, although none (except one, see Model D in Table S3 and Figure S1 in S1 File) were significant at $p < .05$, suggesting an overall higher impact of textual quality on LLM's assessments of conspiratorial narratives. Overall, this suggests that the dictionary-based scoring was less affected by textual quality compared to LLM's scores.

## General Discussion

In this work, we examined how individual worldviews shape the interpretation and communication of narratives. Specifically, we hypothesized that conspiracism would impact the interpretation of ambiguous information in a conspiratorial manner. To test this hypothesis, we conducted a preregistered study in which participants viewed the ambiguous, apocalyptic psychological thriller *Leave the World Behind* and subsequently wrote essays interpreting its meaning. Participants' conspiracism was assessed using the Conspiracy Mentality Scale (CMS), and the level of conspiratorial narrative within

their essays was evaluated using the LLM Claude (by Anthropic). Contrary to our preregistered hypothesis, we found no evidence that conspiracism predicted a conspiratorial interpretation of the film. We thus conducted a second (non-preregistered) study, replacing the CMS with the Generic Conspiracist Beliefs scale (GCB); again, we did not find the expected association.

In a third exploratory analysis, we aggregated data from the two Studies to investigate potential explanations for these unexpected results. We employed an alternative dictionary-based measure to quantify the use of conspiratorial lexicon in participants' essays, which is less dependent on textual quality. Additionally, we applied various indices to assess the textual sophistication and coherence of essays. We found that the conspiratorial lexicon—but not the narrative assessments—correlated significantly with participants' conspiracism and that LLM-assessed conspiratorial narratives were positively associated with measures of textual quality.

In the following sections, we provide an interpretation of why we did not find an association between conspiracism and the emergence of conspiratorial narratives and, in contrast, why the dictionary-based assessment successfully captured associations with conspiracism. We further discuss how the exploratory linguistic markers relate to existing knowledge of conspiratorial language patterns observed online.

### Leave the worldview behind..

Across both Studies 1 and 2, as well as in the aggregated dataset, we found no evidence that participants' conspiracism predicted the narrative conspiratorial level of their essays. In other words, their conspiratorial worldview seemed to be left behind, failing to emerge explicitly in their narrative interpretations. Below, we discuss possible explanations for this null result, considering methodological, motivational, and theoretical factors. We also address the possibility that the expected relationship genuinely does not exist, and we consider broader implications for understanding how CTs may originate.

**The stimulus.** We observed no differences in conspiracism between participants who completed the questionnaire before versus after watching the film. This result aligns with previous research, which found that exposure to an episode of *The X-Files* did not increase participants' conspiracy mentality [140]. As discussed by the authors, the lack of effect might reflect the perceived plausibility or factual grounding of the film (e.g., in contrast to *JFK* [141] or the Chernobyl nuclear power plant explosion [68], which did successfully influence conspiratorial beliefs). Our goal, however, was not to influence or measure participants' attitudes, but rather to observe their sense-making activity. Fictional narratives are routinely used as vehicles for commenting on real-world institutions and power relations [142,143]. As such, regardless of the the perceived realism of the film, conspiratorial interpretations can arise at a literal or allegorical level, or through analogies with what participants perceive as the real world. Thus, if conspiracism shapes interpretation, it should remain detectable in essays about a fictional scenario, even though the openness of the task also allowed for non-conspiratorial readings and may therefore have lowered the base rate of explicitly conspiratorial narratives. Nevertheless, it is also possible that participants interpreted the film as a work of art rather than an account of actual or possible events. In this case, they may have understood the instructions as an invitation to provide a film-criticism-style response, focusing on the director's intentions and overarching message instead of the narrative content per se.

**Participants' motivation.** A lack of participant motivation might also explain our null findings. Although we did not directly measure motivational factors, we indirectly examined participants' motivation through essay length. We did not observe a positive relationship between conspiracism and essays' word count. This is not trivial because prior research consistently shows that conspiracy-related content on websites [57,78–80] and social media platforms (e.g., Reddit; [55]) tends to be longer. Longer texts are generally associated with greater persuasive intent [81].

One plausible explanation for this discrepancy relates to the context of our study. Unlike naturalistic online environments, where individuals are intrinsically motivated to persuade, our participants may have lacked a similar motivation. Online conspiracy promoters typically engage in persuasive communication: even when their personal belief in conspiracies might differ from their spreading behavior, their motivation to disseminate conspiratorial narratives remains strong

[144,145]. In contrast, participants in our study likely viewed the task primarily as a requirement for earning course credit, rather than as an opportunity to persuade. Without the social context or intrinsic desire to persuade others, participants may have been less motivated to elaborate their thoughts or write extensively about conspiratorial themes.

This motivational limitation is consistent with broader challenges in eliciting conspiratorial texts within controlled laboratory environments. Settings such as classrooms or online research platforms (e.g., Prolific, MTurk) typically lack the social dynamics crucial to the formation and propagation of CTs [1,2,146]. In naturalistic settings, such as social media, personal blogs, or conspiracy websites, individuals are strongly motivated to construct and disseminate elaborate conspiratorial content, especially on topics they perceive as personally meaningful. For example, anti-vaccination advocates often create detailed narratives emphasizing perceived dangers, systemic corruption, or threats to personal freedom, driven by the desire to persuade, connect with like-minded individuals, or influence public opinion. In contrast, when participants are asked, as in our lab-based study, to interpret a film for course credit, their motivation likely shifts toward minimal task completion rather than engaging deeply or persuasively with conspiratorial content—even if they hold conspiratorial beliefs.

Finally, it cannot be ruled out that the fear of social stigma often associated with conspiracy beliefs [147] may have played a role among the motivational factors. Our participants were indeed students who were aware that their texts would be eventually read by others. Despite the anonymity of the data collection process, which was meticulously designed to avert such risks, the possibility of self-censorship cannot be discounted [148].

**The prompt.** Beyond motivational factors, another potential explanation for our results relates to the nature of the prompt used to elicit participants' essays (see Prompt S1 in S1 File). The prompt was intentionally designed to trigger cognitive biases associated with conspiracy mentality, known to thrive under conditions of ambiguity and uncertainty [36,37]. However, CTs also fulfill broader epistemic, existential, and social needs [1,2]. Notably, CTs frequently provide explanatory frameworks helping individuals make sense of complex, uncertain, or threatening situations, thus satisfying an epistemic need. It is possible that our prompt, which merely asked participants to "interpret" the film, did not effectively activate this explanatory need. Would our results have differed if the prompt had explicitly engaged participants' epistemic motivations? For instance, rather than asking participants to interpret the film, prompting them to causally explain the depicted events (e.g., *"Why did these events occur?"* or *"Who or what is responsible for these events?"*) might have shifted their responses from subjective interpretation toward structured explanations, potentially eliciting narratives more consistent with conspiracism.

### ...but keep the words

Although participants' conspiratorial worldview did not significantly shape their narratives, we did observe an association between conspiracism and the use of conspiratorial lexicon. In other words, while participants might have left their worldview behind, they retained conspiratorial vocabulary. This finding suggests that conspiracism might emerge more readily at the lexical level rather than through fully structured narratives.

**Cognitive demand.** One plausible explanation involves the cognitive demands associated with narrative construction. The use of familiar or emotionally charged words—particularly those with ideological significance—can occur relatively automatically. In contrast, crafting a coherent narrative involves higher-order cognitive processes such as causal reasoning, temporal sequencing, and thematic integration, all of which are more cognitively demanding [149]. Constructing a narrative necessitates satisfying multiple constraints, including maintaining cohesion and coherence and providing plausible causal explanations for events. Moreover, a high conspiracy mentality does not necessarily imply the ability or motivation to produce fully elaborated conspiracy narratives. Creating such narratives requires advanced causal reasoning and narrative-construction skills that individuals high in conspiracism may not consistently possess or be motivated to apply [120,121]. Thus, participants with higher conspiracy mentality in our study may have used conspiratorial vocabulary (e.g., *deception*, *cover-up*, *elite*) without integrating these words into complete conspiratorial storylines.

**Exposure.** Alternatively, our dictionary-based approach may have picked participants' habitual exposure to conspiratorial discourse. This interpretation aligns with prior evidence indicating higher lexical similarity among conspiratorial webpages across widely disparate topics [58]. Participants might have used recurring idiomatic expressions or lexical patterns (e.g., *"The truth is hidden"*, *"Pulling the strings"* [118]) that, despite not being sufficient individually to form a coherent conspiratorial narrative recognizable by the LLM, still contributed to the conspiratorial lexicon score. Therefore, individuals high in conspiracism may naturally adopt linguistic patterns regularly encountered in online conspiracy communities. Indeed, lexical overlap exists between conspiratorial discourse observed on social media [54,55] and conspiracy-focused websites [57]. Thus, it is reasonable to expect that these lexical patterns could transfer into offline, spontaneous writing, even when participants are neither addressing an audience nor actively trying to persuade.

## Exploratory textual measures

We developed several indices related to textual quality, including lexical and syntactic sophistication, cohesion, and megalalia (the disproportionate use of sophisticated vocabulary). These measures not only served to potentially clarify the LLM's classification outcomes but also provided exploratory insights into whether linguistic patterns found in online conspiratorial discourse manifest in offline texts produced by participants varying in conspiracism. Below, we discuss our findings regarding associations between these linguistic markers and conspiracism in participants' essays, contextualizing our results within existing evidence on online conspiratorial language. In Table 2, we summarize our findings in relation to existent literature.

**Cohesion.** Previous research has found lower cohesion in conspiratorial webpages compared to mainstream ones [58]. Here, however, we observed no such association in participants' short essays. One explanation for lower cohesion in online conspiracy narratives is that they typically jump between multiple unrelated themes, creating highly interconnected but topically scattered narratives [58]. Some scholars have characterized this accumulation of disparate "evidence" (also known as the *mille-feuille* argumentative style or the *Gish gallop*) as a rhetorical strategy designed to overwhelm readers by rapidly presenting multiple arguments without concern for their accuracy or logical coherence [73,150–152]. Therefore, persuasive motivation likely plays a key role in reducing textual cohesion online: to convince others, online conspiratorial narratives strategically connect diverse and unrelated topics, which results in a decreased cohesion.

As previously suggested, our participants may have lacked similar persuasive motivations. The task (writing essays prompted by an ambiguous film), the context (lab study rather than online), or the prompt itself may not have triggered sufficient motivation to persuade. Consequently, participants were unlikely to employ the wide-ranging rhetorical strategies characteristic of online conspiratorial discourse, resulting in similar cohesion levels irrespective of their conspiracy beliefs.

**Table 2. Summary of associations between conspiracism and language features identified in participants' essays in relation to other types of textual sources.**

| Language feature | Essays | Webpages | Social Media |
|---|---|---|---|
| Conspiratorial narrative | = | + | |
| Conspiratorial lexicon | + | + | |
| Word count | = | + | + |
| Cohesion | = | – | |
| Lexical sophistication | – | + | + |
| Megalalia | + | + | |
| Syntactic sophistication | + | + | + |

*Note*. Association between language features and conspiracism: positive (+), negative (-), null (=), or missing / not yet investigated ( ).

Alternatively, the absence of association may be language-specific. Italian is less prevalent online than English, potentially resulting in fewer resources for pre-training Italian embeddings and possibly impacting embedding quality. Additionally, Italian's highly inflectional nature could have influenced the sensitivity of our cohesion metric. Consistent with this interpretation, our validation procedure yielded a larger effect size (the Cohen's $d$) for cohesion in English than in Italian, suggesting potential language-specific limitations.

**Lexical sophistication.**  In previous analyses on LOCO, conspiracy-related texts employed more lexically sophisticated terms compared to mainstream documents [63]. However, in our study, lexical sophistication negatively correlated with participants' conspiracism. This unexpected result might relate to prior research indicating that higher conspiracism often correlates with lower levels of education [121], potentially leading individuals to use less sophisticated vocabulary. However, our participants typically had at least some university-level education, and cohesion was not associated with education levels within our sample, suggesting that this explanation might not fully apply here. An alternative interpretation is that in online conspiracy discourse, sophisticated vocabulary might be employed strategically to signal expertise and authority [73], reflecting persuasive intent. As mentioned above, such motivation was possibly absent among our participants. Without persuasive intents, our participants might not have felt compelled to use sophisticated terminology. This could tentatively explain the observed negative correlation.

**Megalalia.**  Megalalia—the uneven distribution of sophisticated vocabulary—was positively associated with higher conspiracism in both our participants' essays and in LOCO. This preliminary finding suggests that megalalia might be a consistent linguistic feature of conspiratorial discourse, appearing across both online and offline contexts. However, it is somewhat puzzling to observe megalalia in our sample, given participants' presumed lack of clear persuasive motivation. One potential explanation, beyond persuasion, taps into the need for uniqueness, a trait consistently associated with conspiracism (e.g., [72]) and closely linked to personal and social identity [18]. Individuals may thus selectively employ unnecessarily sophisticated vocabulary not only to persuade but also to project an image of exceptional intelligence, competence, or uniqueness [153]. Megalalia might similarly serve to evoke impressions of depth or hidden knowledge, paralleling the pseudo-profound style associated with bullshit receptivity and conspiracism [154]. Consequently, even without overt persuasive intent or an explicitly conspiratorial narrative, individuals with higher conspiracism might still exhibit megalalia as a form of identity signaling or uniqueness-seeking behavior.

**Syntactic sophistication.**  Syntactic sophistication correlated positively with conspiracism in our sample, aligning with previous findings that conspiracy-related accounts on social media [118,119] and conspiracy documents in LOCO also exhibit greater syntactic complexity. Given that a lack of motivation could plausibly explain other unexpected findings (e.g., cohesion), it is intriguing that syntactic sophistication was higher among participants with greater conspiracism. One possibility is that higher syntactic complexity emerges from longer, convoluted sentences characteristic of a stream-of-consciousness style. This interpretation could be consistent with evidence linking conspiracism to thought disorders typically associated with schizophrenia [24,133–135]. However, this interpretation contrasts with findings that individuals with schizophrenia generally exhibit reduced syntactic complexity [155]. Thus, even assuming limited motivation among participants, it remains unclear why syntactic complexity would positively associate with conspiracism and which cognitive or motivational mechanisms might drive this relationship. Impression management, as we discussed for megalalia, could be a tentative explanation.

## Limitations and future directions

In the discussion, we provided an interpretation of why, in our study, conspiracism was associated with conspiratorial lexicon but not with conspiratorial narrative content. In the following, we broaden the focus to additional constraints that were only partially discussed above. In particular, we examine limitations related to our materials, including potential biases in LLM-based classification and the construction of the LOCO and DONALD corpora. Finally, we outline several open

questions intended to guide future research directions. Specifically, we highlight plausible predictors not captured in our design and consider whether conspiratorial narratives may emerge cumulatively.

**LLMs' classification.** A first limitation concerns the prompt we used to elicit LLM judgments, which did not include explicit anchors. Unlike our lexical sophistication norms, where we provided concrete exemplars (e.g., *mum* vs *syphilis*), here neither the models nor human raters were given examples illustrating different levels of conspiratorial strength. In the absence of such anchors, it is difficult to calibrate what counts as a *"very strong conspiratorial narrative"*. Future work relying on LLMs' classification could test whether adding explicit anchors or benchmark texts improves the reliability and validity of these assessments.

A second limitation is inherent to the classification method itself. LLMs may reliably detect conspiratorial narratives primarily when they appear in clearly structured and coherent texts [107,108]. If models are more effective when narratives are cohesive, they may fail to identify conspiratorial content expressed in fragmented, tangential, or loosely organized ways. This interpretation is indirectly supported by our exploratory analyses, which indicated that LLM-based assessments were moderated by indices of textual quality. By contrast, a simpler dictionary measure based on word count successfully captured the relationship between conspiracism and conspiratorial vocabulary use, suggesting that LLM classifications may be more sensitive to narrative coherence than to conspiratorial content alone.

A third additional concern is that LLMs can themselves exhibit a degree of conspiratorial bias [156]. If a model treats conspiratorial claims as relatively plausible, an essay containing such content might receive a lower conspiratorial rating than warranted, resulting in a biased assessment. Such bias, however, should be evenly distributed across the rating continuum and thus systematically decrease conspiratorial ratings for all texts. This would not be problematic unless an artificial floor effect is detected.

Finally, there is a broader concern on how LLMs have been trained. Depending on training material, topics and genres representations are skewed or biased (see, e.g., differences between Google books corpus and British periodicals [157]). This is a well known problem of representability in corpus construction [158], but it can represents a major problem in regard to LLMs, which are often trained on large proprietary corpora whose exact composition is undisclosed, complicating replication and comparability across studies [159].

**LOCO and DONALD's limitations.** Because the conspiracy dictionary used to extract the conspiratorial lexicon was built on LOCO, which also served as a validation corpus for our measures, some aspects of LOCO's construction need to be clarified to flag potential limitations. First, the conspiracy dictionary is not fully independent of the LLM-based assessment. Although it was constructed in a data-driven fashion from LOCO extracting conspiracy-related terms (e.g., *deception*) stripped from event-base ones (*Paris, Diana*), its validation relied on individual documents from DONALD, whose ground truth labels were defined by ChatGPT. This partial overlap may introduce a degree of circularity between feature construction (the dictionary) and the evaluation criterion (LLM judgments).

Second, LOCO was built around events that are known to have generated CTs, gathered from surveys measuring belief in specific CTs [160,161]. However, it could be argued that not all of these events can be unequivocally considered CTs. For example, there is nothing inherently harmful about Bigfoot itself [162], or about the (alleged) faked deaths of popular figures such as Elvis Presley and Princess Diana (or Paul McCartney). Although these events can be considered as facts allegedly known by authorities but hidden from the public (consistent with the definition proposed by [163]), they seem to lack the clear malicious intent that is often taken to characterize CTs [2,3]. This ambiguity in the seed events may limit the content validity of LOCO as a corpus of genuine conspiratorial documents and, by extension, of the dictionary derived from it.

Third, beyond content validity, not all documents in LOCO actually contain CTs, with a portion of documents that mention the event's keyword (e.g., "Sandy Hook") without any conspiratorial discussion of the event itself (the Sandy Hook school shooting) [164]. This reflects a well-known problem in corpus collection [101], namely a trade-off between sample size and accuracy. On the one hand, gathering all documents from a source known to deliver CTs ensures a large sample

size but increases the likelihood of false positives, as it also includes documents that do not mention CTs [164]. On the other hand, false positives are minimized if documents are carefully selected to include CTs, but this results in a costly and time-consuming endeavour, with limited sample size and heterogeneity, and may introduce sample bias if documents are selected based on the same lexical features that are later used to analyze their lexical content. These corpus-level constraints should be kept in mind when interpreting the performance and generalizability of both the dictionary and the measures derived from it.

**Other psychological contributors.** In this work, we focused on conspiracism as a unique predictor of the emergence of conspiratorial narratives. However, it must be acknowledged that other psychological traits may also contribute to the production of such narratives. Below, we highlight several candidates that future research could explicitly incorporate alongside conspiracism.

Schizotypy—a subclinical form of schizophrenia characterized by paranoia, magical ideation, and unconventional thinking—has been identified as one of the strongest psychopathological predictors of conspiracism [24]. Language on the schizo spectrum, including both schizophrenia and schizotypy, has been described as vague, circumstantial, meta-phorical, overelaborate, and stereotyped, with loose semantic associations [165,166], heightened sensitivity to associative and irrelevant stimuli [167,168], reduced semantic cohesion, and deficits in temporal, causal-motivational, and thematic coherence [133–135]. Schizotypy could therefore help explain some narrative and linguistic features frequently observed in CTs, such as high thematic heterogeneity, hyper-interconnectedness, loose semantic associations, and lower topic specificity and lexical cohesion [58–61,63].

Magical thinking, which overlaps with both schizotypy and conspiracism [169], is particularly connected to causality and agency—two core ingredients of conspiracy theorizing [45–47]. This trait may facilitate the endorsement of highly improbable or counterintuitive CTs, such as the belief in omnipotent and omniscient agents (e.g., Reptilian shape-shifters [43]). In the mind of magical thinkers, conspirators may be perceived not only as malevolent but also as virtually flawless in their activity. Even socially marginalized or low-status groups such as immigrants are sometimes portrayed in conspiratorial narratives as possessing disproportionate power to destabilize entire nations, "impose their rules", or orchestrate demographic takeovers. This perceived omnipotence, applied to groups typically seen as relatively power-less, exemplifies the magical and counterintuitive logic underlying many CTs and coherently sustains a narrative of a vast and hidden agenda.

Conspiracy believers are also more susceptible to probabilistic reasoning errors and have a biased conception of ran-domness in which coincidences are seen as suspicious rather than expected [33,42,170–172]. As such, they may disre-gard the probabilistic reality that large-scale conspiracies are prone to failure due to human error and increasing exposure risks as the number of participants grows [173]. One probabilistic error in particular such as the overestimation of the likelihood of co-occurring events—the conjunction fallacy—may be especially relevant for the generation of conspiratorial narratives, as it could help explain the construction of hyperconnected and thematically heterogeneous narratives that bind together otherwise unrelated events.

Another potential predictor of the emergence of conspiratorial narratives is belief in specific CTs. Although conceptu-ally related, correlated, and often used interchangeably by researchers [174], belief in CTs and conspiracy mentality are distinct psychological constructs [22,174–177]. Conspiracy mentality reflects a general tendency to distrust authority and suspect hidden motives, typically assessed with abstract or non-specific statements, and may be more closely associated with general skepticism, political cynicism, and information-seeking motives. Conversely, belief in specific CTs involves endorsement of concrete and often epistemically risky claims and is more strongly linked to pathological cognitive tenden-cies, such as magical thinking, schizotypy, or reduced analytic reasoning [177,178]. Given their more concrete and asser-tive nature, specific CTs may represent a stronger predictor than conspiracy mentality for the generation of conspiratorial narratives, which typically require elaboration, detail, and a degree of belief commitment that aligns more naturally with endorsement of specific, risky claims than with generalized skepticism alone. Our effect sizes support this conjecture: the

relationship between the essays' level of conspiratorial narrative and conspiracism was higher for the more epistemically risky generic conspiracist scale [31] compared to the conspiracy mentality scale [91].

Finally, motivated cognition, partisanship, and affective polarization may also be important contributors. CTs frequently denigrate political or social opponents [53,179,180]; this applies, for example, to QAnon [181–184] and to partisan-oriented CTs about climate change endorsed on both the left and the right [185]. Importantly, CTs can be endorsed or propagated for reasons of convenience [179]. They are not necessarily generated only by sincere believers but can be deliberately constructed and disseminated to achieve strategic goals, such as influencing political behavior or generating profit [144,145]. For instance, many of Donald Trump's conspiratorial claims lack evidentiary support and may represent a type of reasoning that is irrational, internally inconsistent, or performative rather than sincerely believed [186,187]. An important avenue for future research is thus to determine the extent to which CTs are produced intentionally, independent of genuinely held beliefs, versus emerging organically from cognitive biases and dispositional tendencies such as conspiracism, schizotypy, magical thinking, and probabilistic reasoning errors.

**Cumulative effect.** As our data appear to suggest, the actual impact of conspiracism on the emergence of conspiratorial narratives may be relatively small and, as discussed, contingent on other traits and motivations. Given that CTs do exist, how do they originate and propagate? It is possible that individual contributions to conspiratorial discourse might be modest, yet fully fledged CTs could emerge cumulatively [188]. Initial ideas may gradually develop into elaborated narratives through repeated social interactions among multiple individuals over time [59,62,189,190]. The QAnon movement illustrates this process, encapsulating many specific beliefs—even conflicting ones—without a single official version [181,182]; narratives are generated, recombined, and personalized, where individual believers assemble their own conspiratorial "package". Even before QAnon, analyses of the subreddit `r/conspiracy` similarly suggested a decentralized, cumulative pattern of narrative construction [59]. Future research could investigate this cumulative hypothesis by employing paradigms inspired by cultural transmission studies [86,191], allowing researchers to examine how narratives evolve and transform across repeated social exchanges.

**Future directions.** Although our work paves the way for understanding the potential link between conspiracism and language production, several questions remain open, particularly in light of the methodological limitations of our current approach. A recurrent issue in our discussion concerns the extent to which conspiracism overlaps with, or diverges from, persuasive intent. Future experimental studies could systematically manipulate participants' motivation to persuade (e.g., explicitly instructing participants to convince an audience) and examine how persuasive intent interacts with conspiracism. Such studies would help clarify whether the linguistic markers identified here (e.g., megalalia, sophistication) emerge specifically due to persuasive motives, conspiratorial cognition, or a combination of both.

A related open question is whether certain linguistic features are context-dependent—activated only by relevant topics or motivational factors—or whether they reflect stable traits of individuals high in conspiracism across diverse settings. Experimental manipulations of topic relevance or incentives to persuade could shed light on whether conspiratorial language emerges only in specific contexts or consistently characterizes the language production of highly conspiratorial individuals, regardless of situational factors.

More comparative research is also necessary to examine how conspiratorial discourse emerges across different venues (e.g., online webpages, social media platforms, lab-based tasks). Applying consistent analytic approaches across these various contexts—as we did by comparing LOCO documents with our participants' essays—may reveal whether linguistic markers of conspiracism remain stable or differ systematically across communication environments.

Although we did not find evidence that conspiracism leads to overt conspiracy narratives, we observed that participants high in conspiracism used a greater amount of conspiratorial lexicon. This suggests that conspiracism might manifest primarily at the lexical level, rather than through well-structured narratives. If explicit conspiracy narratives do not strongly originate from individuals with high conspiracy beliefs, then who is actually authoring the CTs widely encountered online?

It remains plausible that at least some CTs are strategically constructed—often for political, ideological, or economic gain—rather than arising solely from genuine believers' worldviews and subsequently amplified via social interactions. Future studies employing paradigms such as cultural transmission experiments, collaborative storytelling approaches, or longitudinal analyses could help determine whether conspiratorial narratives emerge cumulatively from individuals' cognitive biases or whether they are predominantly products of intentional persuasion strategies by strategically motivated actors.

## Conclusions

We investigated how conspiracism shapes narrative interpretation and communication by analyzing participants' essays interpreting an ambiguous movie. Contrary to our expectations, individuals with higher conspiratorial beliefs did not produce more explicitly conspiratorial narratives, but they did show greater use of conspiracy-related vocabulary, suggesting that conspiratorial cognition might manifest more readily at a lexical rather than narrative level. Exploratory linguistic analyses revealed that conspiracism was associated with selective use of sophisticated vocabulary (megalalia) and higher syntactic complexity, highlighting parallels between offline and online conspiracy discourse. Future studies should investigate the contexts and motives driving the emergence of conspiratorial narratives, as well as the extent to which conspiracy theories are produced intentionally for strategic purposes, independently of genuine conspiratorial beliefs.

## Supporting information

**S1 File. Prompts used to elicit film interpretations, evaluate Italian essays for conspiratorial content, translate the conspiracy dictionary, generate syntactic complexity corpora, and extract lexical sophistication norms; complete list of Italian dictionary terms used for lexical conspiracism assessment; alternative three-dimensional prompt for evaluating conspiratorial content with correlation matrix comparing metrics; moderated regression analyses examining whether textual quality affects conspiracy content detection; correlation matrices between conspiracy-related variables and narcissism and attachment style measures for Studies 1 and 2; and correlation coefficients between conspiracism, conspiratorial content, and LIWC dictionaries.**
(PDF)

**S2 File. Inclusivity-in-global-research-questionnaire.**
(PDF)

## Acknowledgments

We would like to thank Elisa Valiante, Lorenzo Eugeni, and Lorenzo Picca for their contribution to data collection and coding.

## Author contributions

**Conceptualization:** Ines Adornetti, Francesco Ferretti.

**Data curation:** Alessandro Miani, Daniela Altavilla.

**Formal analysis:** Alessandro Miani, Daniela Altavilla.

**Investigation:** Alessandro Miani, Ines Adornetti, Daniela Altavilla, Valentina Deriu, Alessandra Chiera, Francesco Ferretti.

**Methodology:** Alessandro Miani, Ines Adornetti, Daniela Altavilla, Francesco Ferretti.

**Project administration:** Ines Adornetti, Francesco Ferretti.

**Resources:** Alessandro Miani, Ines Adornetti, Daniela Altavilla, Valentina Deriu, Alessandra Chiera, Francesco Ferretti.

 

**Supervision:** Ines Adornetti, Francesco Ferretti.

**Validation:** Alessandro Miani.

**Writing – original draft:** Alessandro Miani, Ines Adornetti.

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
