## [Decision Letter · Decision Letter 0]

10 Jun 2025

Dear Dr. Miani,

Thank you for submitting your manuscript to PLOS ONE. After careful consideration, we feel that it has merit but does not fully meet PLOS ONE’s publication criteria as it currently stands. Therefore, we invite you to submit a revised version of the manuscript that addresses the points raised during the review process.

We look forward to receiving your revised manuscript.

Kind regards,

Romy Sauvayre, Ph.D., HDR

Academic Editor

PLOS ONE

https://journals.plos.org/plosone/s/file?id=ba62/PLOSOne_formatting_sample_title_authors_affiliations.pdf....

4.  We notice that your supplementary tables, S1, S2 and S3 are included in the manuscript file. Please remove them and upload them with the file type 'Supporting Information'. Please ensure that each Supporting Information file has a legend listed in the manuscript after the references list.

Reviewers' comments:

Reviewer's Responses to Questions

**Comments to the Author**

1. Is the manuscript technically sound, and do the data support the conclusions?

Reviewer #1: Yes

Reviewer #2: Yes

Reviewer #3: Yes

Reviewer #4: Yes

2. Has the statistical analysis been performed appropriately and rigorously?

Reviewer #1: Yes

Reviewer #2: Yes

Reviewer #3: I Don't Know

Reviewer #4: Yes

3. Have the authors made all data underlying the findings in their manuscript fully available?

Reviewer #1: Yes

Reviewer #2: Yes

Reviewer #3: Yes

Reviewer #4: Yes

4. Is the manuscript presented in an intelligible fashion and written in standard English?

Reviewer #1: Yes

Reviewer #2: Yes

Reviewer #3: Yes

Reviewer #4: Yes

Reviewer #1: This is a well-done paper that attempts to answer an important question. Further, no one else, as far as I am aware, have attempted to do this. I support moving towards publication.

Ln. 2 I am not sure that conspiracy theories flourish under uncertainty. The beliefs seem to be pretty stable over time at the mass and individuals levels. So, I am not sure this sentence is needed. The question you really want to address is how these narratives start and who starts them.

I think the paper's intent is a good one - I have yet to see a paper attempt to see if people generated conspiracy theories in response to something. Obviously, people do make up conspiracy theories, they are man made things. Somebody has to make them up. I would like to see the authors set this up more clearly: what are the alternative mechanisms in play? People can make them up or what? They are only made up by insincere people or by people with serious reasoning deficiencies?

We do know, at least anecdotally, that people do come to conspiracy theories on their own intuition. Consider the all of the ancillary conspiracy theories created in the QAnon community. People in that group were making up new things everyday and everyone seemed to have their own set of conspiracy theories.

This gets me to the treatment: This treatment was a Hollywood movie and not a real event. So, I wonder what this really tells us. Having seen the movie, it is weird and its not clear what the cause of the apocalypse is.

I would agree with the authors that it may take more than a conspiracy worldview to create conspiracy theories. It may take some bits of schizotypy and traits like that.

Try to consistent with the language. Pick your terms, define them, and stick with them. I see conspiracism, conspiracy worldview, and conspiracy thinking. Are these the same thing?

The monological discussion, lines 86-94, could be removed. Monological belief systems and conspiracy mentality/worldview, are not the same thing. Both would show correlations between conspiracy theory beliefs, but the underlying mechanisms would be different. In the former, beliefs drive beliefs; in the latter a latent disposition drives beliefs.

While LOCO is a great database and tool, it stretches the definition of conspiracy theory a bit. Bigfoot and Elvis are included in that database, but the existence of bigfoot is not a conspiracy theory (who is conspiring?) and a singer faking his own death (where is the malicious intent or harm?) would not fit most definitions. They are epistemically-suspect beliefs, but they do not allege a conspiracy, necessarily.

I think the interpretation of the results on line 301-305 is correct. The conspiracy mentality construct is just a general disposition, and in itself might not get a person to make up conspiracy theories all the time without other psychological motives involved.

Reviewer #2: Using AI-driven classification and dictionary-based lexical analysis, this manuscript presents an innovative exploration of whether an individual’s conspiratorial worldviews primes the spontaneous production of conspiratorial narratives when interpreting ambiguous stimuli. While the authors do not find a correlation between conspiratorial worldview and narrative structure, they do find linguistic correlates of conspiracy beliefs in the narratives produced by participants; further, they provide an intriguing interpretation of these results in the argument that they may be evidence that the individuals producing conspiracy theories in online discourse may simply have mastered the conventions of the genre rather than being true believers. Despite these strengths, I believe the manuscript would benefit from revision, particularly with respect to the methodological framing of the paper.

ChatGPT is used both to develop the prompt asking participants to producing a narrative interpretation of “Leave the World Behind” and to classify the resulting narratives in the discussion starting on p. 6; while the prompt itself is strong, ChatGPT is known to rely heavily on coherence and wellformedness, which, as the authors themselves note, means that it may miss covert conspiratorial content. Deschrijver (2021) discusses the metapragmatic and metacommunicative nature of conspiracy theories, and may provide useful insight into ways to further refine the instructions given to ChatGPT to help it capture some of the more covert or implicit features of conspiracy theories.

Additionally, the black box nature of ChatGPT’s scoring introduces a potential issue for replication, despite the validation steps provided in the study. The authors might consider adding in an additional layer of manual coding, particularly for texts with low cohesion or unconventional structure, or might note this as a limitation of the study.

The black box nature of ChatGPT is again relevant to the generation of Age of Acquisition (AoA) norms on p. 13. While the authors explain that their use of generative AI here is due to the small pool of existing AoA norms for Italian and provide validation measures for the generated data against the existing dataset, this approach divorces language from speakers and it is unclear how ChatGPT came to those ratings. Łuniewska et al. (2016) find consistency in AoA ratings from a cross-linguistic perspective across different language families - it would be interesting to know how ChatGPT’s ratings compare to data collected from native speakers of languages related to Italian to further support the validation of this approach.

In measuring syntactic sophistication (p. 12), the authors don’t seem to consider the contributions of semantics or specific verb type. Counterfactuals and hypotheticals are often salient structures in conspiracy theories, as are negatively evaluative verbs (Catenaccio, 2022), and adding these considerations into the measure of syntactic sophistication may further enrich these findings.

This paper tackles an important and underexplored question: how do conspiratorial worldviews shape the linguistic construction of narratives? The multi-method approach and large sample size are commendable. However, the analysis would benefit from more engagement with metapragmatic dimensions of discourse and from additional methodological considerations. Addressing these issues would significantly strengthen the paper’s contribution to both conspiracy theory research and the linguistics of belief construction.

Reviewer #3: I see great merit in this study in general, as I agree with the authors that much more research must be done on the rhetoric and content of conspiracy theories rather than looking at psychological priors associated with conspiracism alone. I would like to see some version of this article find its way to publication. That said, I do have a few reservations about the essay in its current form:

1. Can you provide the rationale for selecting Leave the World Behind and for structuring this study around a fictional film? While Sam Esmail’s Mr. Robot series and this film both hint toward fictional conspiracies, neither film purports to represent actual events. A more intuitive film might be a documentary about an unexplained actual phenomenon or event that has yet to surface in existing conspiracy theory discourse. I am also concerned that the film was produced for an Anglophone and specifically American audience, but the respondents in the study would have viewed the film with Italian subtitles presumably. Translation itself is an act of interpretation. Thus, both the original cultural and language context of the film introduce additional variables given that the study was conducted on Italian participants. The authors acknowledge the challenge in moving between conspiratorial discourse in different languages, but they do not really address how the film’s original production might pose problems for their study.

2. The prompt, as the authors correctly acknowledge, asked the participants to interpret the film. Given that it is a film set in a fictional dystopic future, it seems likely the participants will not deploy conspiracy theory narratives as they know it’s a work of art rather than an account of actual events. There are so many possible modes of interpretation—analogical, allegorical, literal, or symptomatic—that could be deployed that may not seem explicitly conspiratorial but that may surface in conspiratorial rhetoric in a different way. Therefore, I am not convinced that prompt would easily yield a response with conspiratorial rhetoric. Perhaps the authors could bolster their case for this address these concerns.

3. LOCO corpus and ChatGPT: I also have reservations about using the LOCO corpus and ChatGPT to perform genre and narrative classification. An LLM, as the authors also acknowledge, is best primed to detect the presence of conspiracy narrative already in circulation. In terms of the LOCO corpus, the article could be strengthened by addressing some of the problems with this corpus and conspiracy theory classification raised in Mompelat et al.’s “How ‘Loco’ is the LOCO Corpus? Annotating the Language of Conspiracy Theories.” Although likely less applicable now, it would also be worthwhile to address the concerns with the way ChatGPT was initially trained on corpora that might miss the mark when it comes to correctly identifying certain genres raised in Bandy and Vincent’s “Addressing ‘Documentation Debt’ in Machine Learning Research.” Precisely because it’s not always clear how these LLMs reach their conclusions, I am wary of relying upon them as a way of classifying narratives they previously haven’t encountered. Thus, this study may say more about the limitations of using LLMs for detecting novel conspiracy theories than it does about the inclination of participants predisposed to conspiracism to craft new conspiracy theory narratives to make sense of ambiguous phenomena or events.

I did, however, find the article’s findings when it came to the use of conspiracy-related vocabulary or syntactic sophistication to be sound because this outcome is not dependent on a particular corpus, LLM, or genre. The authors might consider dispensing with the conspiratorial narrative findings at this stage or qualifying those claims even further, as this part of the article undercuts its strengths.

Reviewer #4: This paper examines an important question—do conspiracy believers spontaneously generate conspiracy narratives when interpreting threatening, uncertain events?—using a highly original method: an analysis of essays discussing the meaning of a movie. The methods, while innovative (e.g., using ChatGPT to rate the conspiracist nature of essays), were subjected to multiple checks to ensure their validity (e.g., having human judges rate the conspiracist nature of the essays and examining how their ratings correlate with ChatGPT scores). The authors also provide interesting additional analyses related to syntactic structure and vocabulary.

I am enthusiastic about this research. I have moderate comments that should be addressed before I can recommend acceptance.

Moderate comments:

• The discussion could be improved. As it stands, it is quite long and somewhat scattered across various interpretations of “why didn’t it work?”. My personal view on null results is that one can always find numerous reasons why an experimental manipulation did not yield the expected outcome. I believe the discussion should focus on theoretically relevant points, particularly the discrepancy between conspiracist narrative ratings (as coherent pieces) and conspiracist vocabulary. I was struck by the weak correlation between these two variables. From my naïve perspective, conspiracist vocabulary should be strongly related to whether a text actually argues for the existence of a conspiracy. This discrepancy deserves deeper exploration and opens up very rich and exciting theoretical questions.

• In the introduction, you tie conspiracism to the need-based approach to conspiracy theories proposed by Douglas et al. (2017) (line 41). However, these are actually two different concepts. Douglas et al.’s (2017) article discusses conspiracy belief and belief in conspiracy theories, and does not mention conspiracy mindset, conspiracism, or conspiracy mentality. Please revise the phrasing in this paragraph to avoid implying that the need-based approach is central to understanding the generic conspiracist mindset.

• While I understand that it does not affect the results, I find the decision to aggregate the two dimensions of the Conspiracy Mentality Scale in Study 1 into a single score questionable. This scale is notable for its attempt to disentangle conspiracist ideation from “healthy suspicion” toward authorities. This issue becomes especially relevant in the discussion between Studies 1 and 2, where some CMS items are described as reflecting an accurate perception of the Italian political system. These items actually belong to the “healthy skepticism” subscale, so it is unsurprising that they do not appear strongly conspiracist. I therefore recommend conducting analyses that include both subdimensions separately (e.g., CT_narrative ~ healthy_skepticism + conspiracist_ideation) in Study 1. The intermediary discussion should be updated accordingly.

Minor comment:

• The abstract begins with the sentence “Conspiracy theories (CTs) tend to flourish under uncertainty.” This phrasing suggests that you manipulated uncertainty, whereas the study examined whether individuals spontaneously mobilize conspiracy narratives to explain ambiguous stories. Consider removing or rephrasing this sentence.

I wish the authors the best of luck with their work.

Kenzo Nera (signed review)

.

Reviewer #1: **Yes:** Joseph UscinskiJoseph UscinskiJoseph UscinskiJoseph Uscinski

Reviewer #2: No

Reviewer #3: No

Reviewer #4: **Yes:** Kenzo NeraKenzo NeraKenzo NeraKenzo Nera

---

## [Decision Letter · Decision Letter 1]

10 Feb 2026

Dear Dr. Miani,

Thank you for submitting your manuscript to PLOS ONE. After careful consideration, we feel that it has merit but does not fully meet PLOS ONE’s publication criteria as it currently stands. Therefore, we invite you to submit a revised version of the manuscript that addresses the points raised during the review process.

This manuscript raises an interesting question. Please address the reviewers' comments. Additionally, there appears to be a slight misalignment between the title and the main findings of the paper. The title suggests an effect of conspiracism on writing, yet the results from the three studies diverge somewhat from this focus. Please clarify the connections and distinctions between related terms such as writing, conspiratorial narratives, conspiratorial discourse, and genuine conspiratorial beliefs.

We look forward to receiving your revised manuscript.

Kind regards,

Haixia Wang

Academic Editor

PLOS One

Journal Requirements:

Reviewers' comments:

Reviewer's Responses to Questions

**Comments to the Author**

Reviewer #1: All comments have been addressed

Reviewer #2: All comments have been addressed

Reviewer #4: All comments have been addressed

2. Is the manuscript technically sound, and do the data support the conclusions?

Reviewer #1: Yes

Reviewer #2: (No Response)

Reviewer #4: Yes

3. Has the statistical analysis been performed appropriately and rigorously?

Reviewer #1: Yes

Reviewer #2: (No Response)

Reviewer #4: Yes

4. Have the authors made all data underlying the findings in their manuscript fully available?

Reviewer #1: Yes

Reviewer #2: (No Response)

Reviewer #4: Yes

5. Is the manuscript presented in an intelligible fashion and written in standard English?

Reviewer #1: Yes

Reviewer #2: (No Response)

Reviewer #4: Yes

Reviewer #1: The authors have addressed my comments appropriately. I appreciate the attention to detail and specific answers to each comment.

Reviewer #2: (No Response)

Reviewer #4: As far as I am concerned, I am satisfied with how the authors addressed my initial comments. At this point, I only have minor comments:

• P4: “we focus on one potential candidate, namely conspiracism, and test whether it is capable of shaping the interpretation and communication of narratives.” This suggests a causal impact of conspiracism on narrative generation. The method does not enable causal conclusions, so consider rephrasing (“we examine if conspiracism is associated with...” (I know it’s a cliché and annoying comment, but I think it is important)).

• P8: “although Claude (r176 = .701, [.617, .769], p < .001) performed better than ChatGPT (r178 = .676, [.588, .748], p < .001)” Since the confidence intervals overlap, I do not think you need to state that Claude performed better than ChatGPT. I’d just state that both LLMs did well.

• p. 16: “being schizotypy the strongest psychopathological predictor of conspiracism” The grammar seems off here.

• p. 18: “As for the interactions, on average, the absolute coefficients of the models for LLM-based assessments were larger than those for the dictionary-based assessment, although none were significant at p < .05, suggesting a higher impact of textual quality on LLMs’ assessments of conspiratorial narratives.”

• In the supplement, there is actually a significant interaction for graph D (the interaction is significant at p = .005). This might be difficult to interpret because the result with Claude is not significant (p = .14), but both are in the same direction, so there might be a meta-analytical effect here. It suggests that the conspiracy-ness of the text is negatively associated with conspiracy mentality when the writing is of high quality, and positive when the writing is of poor quality. You mention this in the limitations.

In the limitations:

• “A third additional concern is that LLMs can themselves exhibit a degree of conspiratorial bias [152]. If a model treats conspiratorial claims as relatively plausible, an essay containing such content might receive a lower conspiratorial rating than warranted, resulting in a biased assessment.” I don’t see this as a real problem, since the rating is continuous. Worst case, the average rating is a little high; it does not necessarily mean that it’ll fuck up your results (except if it results in a ceiling effect that fails to capture high levels of conspiracism).

• I suppose it is in response to a reviewer, but I do not see the necessity to acknowledge that there might be other processes contributing to the emergence of conspiracy narratives (magical thinking, etc.), especially given the length of the limitations section. Of course, there are other processes involved; it’s the same for every possible dependent variable in social psychology. That said, the “conspiracy as a cumulative work” hypothesis is really cool.

Kenzo Nera (signed review)

.

Reviewer #1: No

Reviewer #2: No

Reviewer #4: **Yes:** Kenzo NeraKenzo NeraKenzo NeraKenzo Nera

---

## [Editor Report · Decision Letter 2]

19 Mar 2026

Leave the world(view) behind, but keep the words: The effect of conspiracism on writing

PONE-D-25-17874R2

Dear Dr. Miani,

We’re pleased to inform you that your manuscript has been judged scientifically suitable for publication and will be formally accepted for publication once it meets all outstanding technical requirements.

Kind regards,

Haixia Wang

Academic Editor

PLOS One
---

## [Editor Report · Acceptance letter]

PONE-D-25-17874R2

PLOS One

Dear Dr. Miani,

I'm pleased to inform you that your manuscript has been deemed suitable for publication in PLOS One. Congratulations! Your manuscript is now being handed over to our production team.

Kind regards,

on behalf of

Dr. Haixia Wang

Academic Editor

PLOS One